# The Pathogen *Aeromonas salmonicida achromogenes* Induces Fast Immune and Microbiota Modifications in Rainbow Trout

**DOI:** 10.3390/microorganisms11020539

**Published:** 2023-02-20

**Authors:** Baptiste Redivo, Nicolas Derôme, Patrick Kestemont, Valérie Cornet

**Affiliations:** 1Research Unit in Environmental and Evolutionary Biology (URBE), Institute of Life, Earth & Environment (ILEE), University of Namur, 5000 Namur, Belgium; 2Département de Biologie, Institut de Biologie Intégrative et des Systèmes (IBIS), Université Laval, Québec, QC G1V 0A6, Canada

**Keywords:** core microbiota dysbiosis, bacterial challenge, aquaculture, fish health

## Abstract

Environmental stressors can disrupt the relationship between the microbiota and the host and lead to the loss of its functions. Among them, bacterial infection caused by *Aeromonas salmonicida*, the causative agent of furunculosis, results in high mortality in salmonid aquaculture. Here, rainbow trout were exposed to *A. salmonicida achromogenes* and its effects on the taxonomic composition and structure of the microbiota was assessed on different epithelia (gills, skin, and caudal fin) at 6 and 72 h post-infection (hpi) using the V1–V3 region of the 16S rRNA sequencing. Moreover, the infection by the pathogen and immune gene responses were evaluated in the head kidney by qPCR. Our results suggested that α-diversity was highly diverse but predominated by a few taxa while β-diversity was affected very early by infection in the gills after 6 h, subsequently affecting the microbiota of the skin and caudal fin. A dysbiosis of the microbiota and an increase in genera known to be opportunistic pathogens (*Aeromonas*, *Pseudomonas*) were also identified. Furthermore, an increase in pro-inflammatory cytokines and virulence protein array (*vapa*) was observed in trout head kidney as soon as 6 hpi and remained elevated until 72 hpi, while the anti-inflammatory genes seemed repressed. This study suggests that the infection by *A. salmonicida achromogenes* can alter fish microbiota of gills in the few hours post-infection. This result can be useful to develop a non-invasive technique to prevent disease outbreak in aquaculture.

## 1. Introduction

Epidermal surfaces of metazoans are the interfaces between the host and the environment and are recovered by mucus layers. These mucus layers are colonized by microorganisms that exploit these particular ecological niches and provide many benefits for the host such as fish, where the microbiota is known to provide a protection against pathogen infection [1]. Indeed, this microbiota is involved in the maturation of innate and adaptive immunity, its stimulation, and the defense against pathogens by avoiding their colonization, acting as passive/non-passive barrier [2,3,4]. Moreover, in rainbow trout (*Oncorhynchus mykiss*) skin, the commensal lactic bacteria were shown to prevent the colonization of *Lactococcus garvieae*, etiological agent of lactococcosis [5], by occupying the ecological niches of this latter one [6]. In salmonids, Robertson et al. highlighted that the bacteria from the *Carnobacterium* genus were able to inhibit the growth and the proliferation of several pathogenic strains such as *Aeromonas hydrophila*, and *Aeromonas salmonicida* [7]. Boutin et al. (2012) showed that indigenous bacteria from unstressed brook charr (*Salvelinus fontinalis*) skin can exert an antagonistic effect against *Flavobacterium* infection [8]. Therefore, it is essential for the host to maintain a successful communication with its microbiota through appropriate immunity. However, this tight cross-talk can be disrupted by environmental factors such as the diet, pollutants, or infection [9] and lead to a dysbiosis. Dysbiosis is defined as a disturbance in microbial composition or a disruption in communication between the bacterial community and the host [10]. Many opportunistic pathogens compete with commensal bacteria to make the most of the food source on the epithelial surfaces [11]. Two strategies have emerged in bacteria to make the most of this particular ecological niche. On the one hand, some bacteria have developed invasive mechanisms such as parasitism or pathogenicity to gain access to more host resources while the fish’s immune system acts to stop the infection (red queen concept). On the other hand, some bacteria have co-evolved with the mucosal immune system in mutualistic or synergetic ways by using host by-products which form the microbiota [12]. When facing an opportunistic pathogen, the host is not the only component that undergoes changes but it is the whole relationship that is modified [13]. The infection modifies the immune status of the fish that tries to cope with the infection by the secretion of antibacterial and pro-inflammatory compounds [14,15]. Pathogens invade the mucus layer and eliminate certain bacteria from the microbiota to grow. Usually, these modifications are followed by an increase in opportunistic pathogens that profit of the immune depression of the fish [16]. Several studies have shown that opportunistic pathogens adhere to and pass through the mucus layers of various organs such as the gills, skin, lateral line, and gastrointestinal tract [17,18]. The gills have a large surface area for exchange with the surrounding water for oxygen uptake and direct and rapid access to the capillaries for oxygen delivery to all organs. Although the skin and, to a lesser extent, the caudal fin have a large surface area, the number of cells that must be passed through before reaching the vascular system is much greater. Because of their histological structure and their role in respiration, the gills are considered to be the main entry point for bacterial infections in fish.

Rainbow trout is one of the major freshwater fish species cultivated worldwide. However, it is susceptible to many types of disease agents including bacteria, viruses, and parasites. Among the known pathogens, *Aeromonas salmonicida achromogenes* is responsible for furunculosis, which is characterized by boils and systemic hemorrhaging, resulting in significant mortality in aquaculture systems [19,20] and consequent financial losses. Moreover, several studies have demonstrated the ability of *A. salmonicida achromogenes* to trigger inflammation and immune processes in rainbow trout and zebrafish [21,22,23]. Nevertheless, very little is known about the effects of this infection on the rainbow trout microbiota and immune status modifications associated. In this study, we aimed to evaluate the evolution of both commensal microbiota and immune responses of rainbow trout over time induced by the bath infection with the pathogen *A. salmonicida achromogenes*. This bacterial infection provides a better understanding of how the different microbiota of the rainbow trout are affected depending on the epithelial locations (skin, gills, and caudal fin). As proposed by Slinger et al. [24] for salmon, the methodology used in this study can be adapted to develop a non-invasive technique to monitor fish health in aquaculture and anticipate the onset of disease due to bacterial infection.

## 2. Material and Methods

### 2.1. Fish Rearing

Bacterial challenge and fish sampling were approved by the local Ethic Committee for Animal Research of the University of Namur, Belgium (Protocol number: 16272 KE). Rainbow trout juveniles were obtained from the commercial Hatrival pisciculture (Hatrival, Belgium) and were randomly distributed into 100 L glass tanks (20 fish/tank) in a recirculating system. The average fish length was 19.7 ± 3.5 cm from head to tail and the average weight was 92.86 g ± 22.86 g. Fish were allowed to acclimate to the new standard conditions for 1 month, i.e., 13.5 °C, 8 mg/L of O_2_, with a control level of ammonium and nitrite (respectively < 0.75 and 0.01 mL) and under a photoperiod of 12Light:12Dark (L:D). Fish were fed twice a day 6 days per week with commercial food (Coppens, Supreme-22). No mortalities were reported during this acclimatization.

### 2.2. In Vivo Experiment: Bacterial Challenge and Fish Sampling

The *Aeromonas salmonicida achromogenes* strain used for the infection was provided by the Centre d’Economie Rurale (CER-group, Marloie, Belgium) and was originally isolated from an infected fish that exhibited furunculosis. The strain was identified as a *A. salmonicida achromogenes* using biolog system and was deposited in the BCCM collection (Belgian coordinated collection of microorganisms) with code LMG p-31558. The virulence and LD50 was assessed during a preliminary test infection, and various bacterial doses were performed to determine the LD50 CFU of the targeted rainbow trout population (LD50 = 3.1 × 107 CFU/100 g fish body weight) [25]. Bacteria were isolated from a unique colony in sterile brain heart infusion (BHI) solid medium (Sigma Aldrich, Saint-Louis, MO, USA) and then incubated at 22 °C for 18 h. Bacteria were cultured until reaching 3.1 × 10^9^ CFU/mL just before the start of the experiment. Prior to this bath infection, 10 fish were euthanized with an overdose of buffered MS222 (200 mg/L) and gills, caudal fin, and skin mucus were sampled with individual swabs (Copan, Italy) to perform microbiota analyses and the head kidney were kept for immune gene expression measurement. Then, fish were exposed to the *A. salmonicida achromogenes* strain for 1 h by bath infection at a final concentration of 10^6^ CFU/mL. Afterwards, the exposed fish were rinsed with clean water and placed in glass tanks (20/tank) corresponding to the different sampling time points (6, 24 and 72 hpi post-infection) to avoid disruption of the microbiota that might be due to sampling at the previous time. As the sampling at different timepoints can induce a stress due net chasing that in turn would potentially affect the different analyzed microbiota, only one tank per timepoint was used. Because each tank was part of a recirculating system, water from each tank was collected through a single filter and returned to the entire system, leading to identical water in each tank. The analysis of water commensal microbiota did not show any significant differences in microbial diversity confirming the absence of a tank effect between timepoints. Using the same protocol as for fish euthanized before bacterial infection, ten fish were randomly sampled at 6 and 24 h post-infection (hpi); and 6 fish were sampled at 72 hpi. Microbial swabs and tissues were stored at −80 °C prior to DNA extraction and mRNA extraction. For each timepoint, water samples (500 mL) were also collected in triplicate, centrifuged during 30 min at 21000 g, and the pellets were stored at −80 °C to ensure that changes in individual microbiota were not due to changes in the bacterial community in the water after the previous bath infection. A control group that was not exposed to the bacteria was also conducted on three rainbow trout (skin and gill mucus) to follow the changes that might occur naturally in the different microbiotas within 72 h. All along the experiment, fish welfare was evaluated and fish that had reached the human acceptable endpoints had to be euthanized. Particularly, fish exerting several symptoms and an advanced furunculosis were euthanized. All sampled fish were tested for the presence of *A. salmonicida achromogenes* using expression of its specific *vapa* gene in head kidney samples.

### 2.3. DNA Extraction, PCR and 16S Amplicon Sequencing

DNA extractions were performed using a modified protocol of QIAGEN DNeasy Blood and Tissue Kit (Hombrechtikon, Switzerland) for swabs and pellets from water samples. The protocol developed by the manufacturers was used using twice the suggested solution volumes (except for the elution steps) and by introducing a lysis step with lysozyme at 20 mg/mL (30 min at 37 °C) prior to proteinase K lysis step (56 °C overnight). DNA was eluted in two steps in 50 µL of DNAse-free water. Samples were quantified using the Qubit method prior to the first PCR step. The V1-V3 region of the 16S rRNA was amplified using the V1 (27F: 5′-AGAGTTTGATCMTGGCTCAG-3′) forward and the V3 (534R: 5′-ATTACCGCGGCTGCTGG-3′) reverse primer which contained an overhang adapter for the second PCR step. PCR products synthesis was performed using thermocycling with 2.5 μL of genomic DNA, 5 μL of amplicon PCR forward primer (1 μM), 5 μL of amplicon PCR reverse primer (5 μM), and 12.5 μL of 2× KAPA HiFi HotStart Ready Mix (Kapa Biosystems, Wilmington, MA, USA) starting by performing an initial step at 95 °C for 3 min, then 25 cycles of 95 °C for 30 s, 55 °C for 30 s, and 72 °C for 30 s, and a final extension step at 72 °C during 5 min. PCR amplicons were subsequently purified using the MSB Spin PCRapace kit (Invitek, Berlin, Germany) and the concentration was checked using Qubit protocol. Then samples were sent to GenomicsCore (KU Leuven, Belgium) in order to attach the dual index and Illumina sequencing adapters (San Diego, CA, USA) using the Nextera XT index kit (Juno Beach, FL, USA). Second PCR product synthesis was performed using thermocycling step with 5 µL of DNA, 5 µL of Nextera XT Index Primer 1, 5 µL of Nextera Index Primer 2, 10 µL of PCR grade water, and 25 µL of 2× KAPA HiFi HotStart Ready Mix starting by performing an initial step at 95 °C for 3 min, then 8 cycles of 95 °C for 30 s, 55 °C for 30 s, and 72 °C for 30 s, and a final extension step at 72 °C during 5 min. PCR products were purified using AMPure XP beads. Sequencing was performed using Illumina HiSeq 2500 (250 + 50 bp).

### 2.4. Bioinformatics Pipeline and Microbiota Analysis

The sequencing process generated 34,309,182 reads. The sequence data were processed using different independent software. Firstly, low quality sequences were removed, i.e., no ambiguous base, a length varying between 400 and 580 bp, minimum overlap of 15 bp, and a quality threshold set up at 0.8 out of 1 using Pandaseq software 2.10. After this process, 30,100,580 reads were obtained. Then, sequences were processed using 1.9.1 Quantitative Insights Into Microbial Ecology 1.9.1 (QIME 1.9.1) pipeline [26]. Moreover, de-noising and chimera detection were performed prior to clustering step using USEARCH [27] and at the end of this step, 29,046,778 reads were acquired. Afterwards, the sequences were clustered with 97% sequence similarity using UCLUST in a de novo way. The representative sequence for each cluster provided a taxonomic assignment using the SILVA database, aligned on the representative set with PyNAST [28] and finally a phylogenetic tree was built with FastTree. The microbiota analyses were carried out with Phyloseq [29], a package on R. Prior to any analysis, our data were transformed using CSS normalization to reduce the effects of a heterogeneous library size using the metagenomSeq package [30]. For α-diversity, the Chao1 (richness estimator), Shannon and Inverse Simpson indexes were calculated with Phyloseq tools and generalized linear models (GLMs) were used to highlight statistical differences. The microbiome package was used to analyze the core microbiota and this method allowed to obtain a good representation of the different microbiota analyzed. The core microbiota was defined as the bacteria that are present in all the samples from the same tissue but one (present in 90% of the samples for each group), with a relative abundance of at least 0.5%. For β-diversity analyses, Permanova was performed for testing differences between groups of samples with vegan package [31]. Bray–Curtis [32] and Weighted Unifrac distances were used as the quantitative method and Unweighted Unifrac for the qualitative indices [33]. Prior to Permanova, “rare” OTUs from each sample were removed (<0.1%) in order to reduce noise from the dataset [34]. Nonmetric multidimensional scaling (NMDS) was used to visualize differences in the bacterial community based on the same distances discussed above. Negative binomial GLMs were applied to determine the OTU that were differentially abundant across tissues throughout bacterial infection. For these GLMs, we trimmed OTU that were not present in at least 75% of the samples within the experimental group and with an abundance > 0.1%. Models and *p* values for the multivariate generalized linear models were obtained using mvabund (r package) [35] and corrected by the likelihood ratio test for multivariate analysis. The inferences were performed using 999 bootstrap resamplings. The correlation network analysis was achieved in R based on correlations between the abundance of the OTUs for each microbiota separately. Significant Spearman correlations (|r| > 0.75 and corrected *p* value < 0.001) were calculated in the R packages Hmisc, pscych, and igraph. Network layouts are based on the Fruchterman–Reingold process which helps to create clusters since this method tends to place correlated OTUs next to each other.

The functional and metabolic profiles from our 16S amplicons were also predicted using Tax4Fun [36] at the most accurate level. Then, Permanova with Bray–Curtis distance were performed to detect differences between the different experimental groups.

### 2.5. Immune Gene Expression Analysis

The immune response of fish throughout the bacterial infection was assessed using immune gene expression from the head kidney. Total RNA was extracted from the head kidney using Tri-Reagent solution (Ambion, Thermofisher Scientific, Waltham, MA, USA) according to the manufacturer’s instructions. The pellet was dried and resuspended in 100 µL of RNase-free water. Total RNA concentration was determined by NanoDrop-2000 spectrophotometer (Thermo Scientific) and the integrity was checked using the bioanalyzer 2100 (Agilent, Santa Clara, CA, USA) and on agarose gel (1%). Genomic DNA was digested for 15 min at 37 °C with 1U of rDNAse I (Thermofischer Scientific). Then, 1 µg of total RNA was reverse-transcribed using RevertAid RT kit according to the manufacturer’s instructions (Thermofischer Scientific).

Relative expression of 9 coding transcripts were assessed by qPCR. Two housekeeping genes (β-actin and 18S rRNA) were used to normalize the expression of the target genes. Bestkeeper results showed a coefficient of correlation of 0.926 (*p* < 0.001), standard deviation of 0.19 Cq and 0.941 (*p* < 0.001), and standard deviation of 0.37 Cq for 18S RNA, and β-actin [37]. The gene expression of pro-inflammatory cytokines (*il-1β*, *tnfα*), antibacterial proteins (lysozyme (*lyso*), C3 protein of complement system (*c3-4*)), myeloperoxidase from neutrophils (*mpo*), and anti-inflammatory response (*il-10*, *tgfβ*) was evaluated. In addition, the relative expression of the transcript Virulence array protein A (*vapa*), which is involved in the infection and virulence of *A. salmonicida* was measured. The list of specific primers used for gene expression analysis is described in Table 1. qPCR reactions were carried out with SsoAdvanced™ Universal SYBR^®^ Green Supermix (Bio-Rad Laboratories, Hercules, CA, USA) using a 1:100 dilution of the cDNA for target genes and reference genes. Primers for target genes were used at a final concentration of 0.5 µM. The thermal conditions used were 3 min at 95 °C for the initial step, followed by 40 cycles at 95 °C for 30 s and 60 °C for 30 s. qPCR analyses were performed with a StepOnePlus device (Applied Biosystems). The relative gene expression was calculated according to the relative standard curve method based on the geometric mean of the housekeeping genes [38]. Mean Ct results are available in the Appendix A.

### 2.6. Statistical Analysis

All statistical analyses for the immune assessments were performed in R using GLM with the appropriate distribution in order to obtain the normality and homoscedasticity of residuals. We tested the effect of the mucus location, the time after the bath infection, and both combined. Post-hoc analyses were performed using the *multcomp* package using the Tukey test.

## 3. Results

### 3.1. In Vivo Experiment

The health of trout exposed to *A. salmonicida achromogenes* from each pond was monitored throughout the experiment. Several signs of hemorrhages at the base of the fins, small ulcers on the side of the fish, and mortalities were observed. These symptoms are characteristic of furunculosis induced by this pathogen (see Baset (2022) for more information on the description and visualization of the symptoms). No symptoms of disease or mortality were observed on the control fish.

The results of skin and gill microbiota confirmed a stability of the microbiota overtime, meaning that the alpha diversity (Shannon index–*p* value = 0.234; Inverse Simpson–*p* value = 0.31106; Chao1: *p* value = 0.36) and beta diversity indices (Weighted Unifrac: F. Model = 0.62153; R^2^ = 0.17162; *p* value = 0.907; Unweighted Unifrac: F. Model = 1.0028; R^2^ = 0.25052; *p* value = 0.455–Bray–Curtis: F. Model = 0.92073; R^2^ = 0.23484; *p* value = 0.634) do not shift between the timepoints. Moreover, our results indicated that the alpha diversity (Shannon index–*p* value = 7.66 × 10^−7^; Inverse Simpson–*p* value = 0.00107; Chao1: *p* value = 8.41 × 10^−5^) and the beta diversity (Weighted Unifrac: F. Model = 14.9843; R^2^ = 0.48610; *p* value = 0.001–Unweighted Unifrac: F. Model = 5.2763; R^2^ = 0.24644; *p* value = 0.001–Bray–Curtis: F. Model = 15.3811; R^2^ = 0.48788; *p* value = 0.001) were only influenced by the location (skin and gills).

### 3.2. Bioinformatics Analysis

After quality trimming, chimera removal and OTU’s assignments filtering, several sequences varying between 150,000 and 672,000 were obtained. These sequences were clustered in 3022 OTUs, 313 genus, 191 family, 103 order, 52 class, and 30 phyla.

### 3.3. α-Diversity Analysis

The evenness of the different microbiota was calculated through Shannon and Inverse Simpson indexes and the richness was estimated through Chao1 index and are indicated in Table 2. α-diversity measurements (Shannon, Inverse Simpson, and Chao1) were evaluated using GLMs with a Gaussian distribution. Shannon and Inverse Simpson index GLMs revealed that the bacterial infection did not influence α-diversity but that each mucosal epithelium was different depending on the measurement used (respectively, *p* value < 0.001 for Shannon index Inverse Simpson index). Shannon measurements indicated that every microbiota is different from each other with the water with the higher diversity followed by the mucus of the gills, then the mucus of the fin, and finally the mucus of the skin. Whereas the Inverse Simpson index showed that the mucus of the gills and the water indices were not different. The microbiota of the gill mucosa and the caudal fin are not significantly different. Chao1 richness measurements did not highlight any significant effect (*p* value < 0.05) but the results follow the same pattern as observed with the Shannon and Inverse Simpson indexes.

### 3.4. Structure of the Rainbow Trout Core Microbiota throughout Bacterial Infection

The composition of the microbiota during bacterial infection was analyzed (Figure 1). The core microbiota of water, gill mucus, skin, and caudal fin throughout infection were 61.5, 61.8, and 59.4% (pre-infection-6 h-72 h) respectively; 50.3, 55.9, and 69.5% (pre-infection-6 h-72 h), respectively; 92.8, 92.7, and 92% (pre-infection-6 h-72 h), respectively; 76.9, 79.2, and 80.7% (pre-infection-6 h-72 h), respectively, of the OTUs present in their microbiota. The core bacterial community of the water did not seem to be impacted by the bacterial infection except for the *Flavobacteriaceae flavobacterium* which slightly increased after 72 h post-infection.

Gills microbiota appeared directly impacted by the infection since an increase in *Burkholderiaceae polynucleobacter* and of *Pseudomonaceae pseudomonas* was observed 6 h after infection. Moreover at 72 hpi, *F. flavobacterium* rose to 45% of the core microbiota composition. At the opposite, a decrease in *Comamonadaceae* as well as *Burkholderiaceae polynucleobacter* (from 21 to 13% for the *Comamonadaceae* and from 46 to 13% for the *B. polynucleobacter*) was highlighted.

Regarding the core microbiota of the skin, we could not observe any impact of the bacterial exposure during the first 6 h even though there was a slight increase in *B. polynucleobacter* population coupled with a decrease in *F. flavobacterium*. Moreover, the core microbiota of the mucosal skin was much more impacted 72 h post-infection. We observed a large decline in *B. polynucleobacter* population along with a huge increase in *F. flavobacterium* population. We also highlighted an important increase in *Oxalobacteraceae undibacterium* population even though the percentage was quite low.

The mucosal fin core microbiota appears to show a similar pattern to that observed for the skin core microbiota, i.e., there is no real difference in the first 6 h and then it undergoes a significant impact 72 hpi. However, the decrease in *B. polynucleobacter* population seemed to be correlated with an increase in *Comamonadaceae* population. An increase in *Chromatiaceae rheinheimera*, *P. pseudomonas*, and *O. undibacterium* populations was also noted 72 hpi. Despite the bacterial infection by *A. salmonicida achromogenes*, the presence of this bacteria within the core microbiota of the different epithelia was not identified.

### 3.5. β-Diversity Analysis

Phylogenetic diversity (weighted Unifrac), phylogenetic richness (Unweighted Unifrac), and Bray–Curtis diversity were also assessed to compare the bacterial communities. Permanova using these indexes indicated a strong effect of the mucus location as well as the bacterial infection by *A. salmonicida* (Mucus_location*TimepostInfection-Weighted Unifrac: F.stat: 3.813; R^2^: 0;0924; *p* value: 0.001. Unweighted Unifrac: F.stat: 1.3697; R^2^: 0.06384; *p* value: 0.001. Bray–Curtis: F. stat: 3.888; R^2^: 0.1055; *p* value: 0.001). Therefore, these effects were explored through pairwise comparisons Permanova using a Benjamini–Hochberg correction [35] which controls the false discovery rate. All the results are presented in Appendix A. Bray–Curtis and Weighted Unifrac Permanovas suggested that the mucus location has a strong influence on the bacterial communities. This showed that the gills and skin microbiota at 6 and 72 h post-infection were significantly different from the mucus before the infection, whereas fin microbiota was only modified 72 hpi. Furthermore, pairwise comparisons using Unweighted Unifrac distance suggested that there was no significant difference between the different mucus locations but there was a strong effect of the bacterial infection on the bacterial communities. These analyses have been associated with NMDS plots showing strong clustering as observed with the Permanova (Bray–Curtis: Figure 2a; Weighted Unifrac: Figure 2b; Unweighted Unifrac: Figure 2c).

Differentially abundant taxa and correlation network were used to detect OTU variations caused by the bacterial infection. For these GLM, OTUs that were not present in at least 75% of the samples were trimmed. After this filtration, 176, 218, and 141 OTUs for the gills, skin, and fin microbiotas, respectively, were obtained. Among those OTUs, 44 were shared across all microbiota. Models and *p* value for the multivariate generalized linear models were obtained using mvabund (r package) [32] and corrected using the likelihood ratio test for multivariate analysis. The overall microbial composition showed a significant impact of the bacterial infection (analysis of deviance table for the gills microbiota: Dev: 2389; Pr (>Dev): 0.001. Analysis of Deviance Table for the skin microbiota: Dev: 698.4; Pr (>Dev): 0.001. Analysis of Deviance Table for the fin microbiota: Dev: 1110; Pr (>Dev): 0.03). Then, multivariate GLM for each location highlighted a difference induced by the bacterial infection. For gill microbiota, 34 OTUs varied across time while only 16 OTUs varied for both the fin and the skin microbiotas. All the significantly varying OTUs are summarized in the Appendix A. Among the bacteria that have been impacted, three are shared by all the different microbiota locations (denovo 802505 *O. Undibacterium*, 585527 *Comamonadaceae* sp., and 2550725 *C. Rheinheimera*). Nine OTUs are shared between the gills and skin microbiota (denovo 3075002, 802505, 376261, 585527, 2550725, 786594, 1484569, 1638662, and 2178738), six are shared between the gills and fin microbiota (denovo 2455659, 802505, 585527, 2594052, 2550725, and 115553), and five are shared (denovo 802505, 585527, 2550725, 1514040 and 60891). Interestingly, six OTUs from the gills core microbiota (denovo 2455659, 376261, 585527, 115553, 1638662 and 2135443), four OTUs from the skin core microbiota (denovo 802505, 376261, 786594 and 1638662), and five OTUs from the caudal fin core microbiota (denovo 2455659, 802505, 2594052, 115553 and 955978) were impacted by the bath infection. The taxonomy related to these OTUs can be seen in the Figure 3 at the heatmap level.

A correlation network approach was also developed to assess the dysbiosis induced by bath infection for each microbiota using the time post-infection as variable. For the correlated gill network (Appendix A), three complex networks can be distinguished, with the largest in the center. This large network seems to be symmetrically divided into two subnetworks by bacteria that negatively influence the other subnetworks. Among these bacteria, some were found to be modified by the bath infection (*C. rheinheimera*, *X. arenimonas*, *Rhizobiaceae*, *M. mycobacterium*, and *Saprospiraceae*). To this extent, when the abundance of these bacteria is altered, we expect the other part of the network to be altered in the opposite way. However, this is not the case and thus the bacterial challenge has created an imbalance in the composition and correlation of the microbiota. At the bottom, a smaller network contains several putative pathogenic bacteria (Pseudomonas, Acinetobacter, and Flavobacterium) and some of the bacteria present in this network increased dramatically after the bacterial infection.

For the correlated skin network (Appendix A), one big network can be distinguished and is mostly constituted by *Polynucleobacter* bacteria (110 out of the 184 OTUs present in this sub-network) and most of the edges represent positively correlated bacteria. Furthermore, the bacteria that were significantly changed following bacterial infection belonged only to this subnetwork. The other networks did not show any changes in bacteria following bath infection.

For the correlated caudal fin network (Appendix A), four complex networks were observed. Among these four complex networks, three were slightly modified after the bacterial challenge.

The impact of the bath infection on the bacterial network for every mucus location was also calculated. The number of edges (bacterial correlations) linked to each node (OTU) that were changed due to bacterial infection were counted. Then this number was divided by the total number of correlations in the network. A ratio corresponding to the percentage of the bacterial network that was impacted by *A. salmonicida* was obtained and was determined as the dysbiosis ratio. For the bacterial network of the gill microbiota, the dysbiosis ratio is 14.53% (42 edges impacted out of 289). For the bacterial network of the skin microbiota, the dysbiosis ratio is 23.95% (358 edges impacted out of 1495). For the bacterial network of the fin microbiota, the dysbiosis ratio is the lowest (171 edges impacted out of 275).

### 3.6. Functional Analysis through Tax4FUN

Functional profiles of the different microbiota were assessed using Tax4Fun [33], with 65.3% (±25.5%) of all 16S rRNA being mapped to KEGG organisms into the analyses. A total of 6324 KEGG pathways at the highest level were found then regrouped into 11 groups of basic KEGG pathways. Permanova highlighted a significant effect of the interaction between the time post-infection and the mucus location (*p* value < 0.001) and is illustrated by NMDS in the Figure 4a. Therefore, we explored this interaction effect through pairwise comparison Permanova using Bray–Curtis distance with Benjamini–Hochberg correction (Appendix A). The results and *p* value associated are presented in Appendix A. The functional profiles of the water, the caudal fin, and the skin were not impacted by the bacterial infection of *A. salmonicida*. Moreover, the functional profiles of the gill microbiota differed and were strongly influenced by changes 6 h after infection. Moreover, the functional profiles of all the bacterial communities were different from each other in the group before the infection, 6 and 72 h post-infection (except for the fin 6 h vs. water 6 h and fin 72 h vs. gills 72 h comparisons; *p* value > 0.05). Therefore, statistical analyses using GLMs were used and showed that out of the 11 functions found in the gill microbiota, 10 were impacted by the bacterial infection (Amino acid metabolism, Carbohydrate metabolism, Energy metabolism, Glycan biosynthesis/metabolism, Lipid metabolism, Metabolism of terpenoids and polyketides, Nucleotide metabolism, and Xenobiotics biodegradation/metabolism pathways (Figure 4b). GLMs were also used to determine which functions are different between different microbiota and at different post-infection times (Figure 4c). GLMs highlighted that for the group before the infection and 6 h post-infection group, the functional profiles are different between all the microbiota and only 9 for the 72 hpi (except Biosynthesis of secondary metabolites and Metabolism of other amino acids pathways).

### 3.7. Evaluation of the Immune Status of the Fish

First, the virulence of the bacterial strain used was assessed using *vapa* gene expression and the result is presented in Figure 5. The expression level of *vapa* was significantly higher in fish from the 6 h, 24 h, and 72 hpi groups compared with the pre-infection group where no expression was detected (*p* value < 0.001). The expression levels of pro-inflammatory (*il1b* and *tnfα*), anti-inflammatory (*il10* and *tgfβ*), and antimicrobial compounds (*c3*, *mpo*, and *lyso*) coding transcripts are presented in Figure 5. *il1β* gene expression was significantly enhanced in fish that were exposed to the bacteria 24 and 72 h before. For *tnfa*, its expression increased significantly 6 hpi compared with the group before infection but was already decreasing after 24 hpi as this group was no more different from the group before infection (*p* value < 0.01). For the anti-inflammatory gene expressions, there were no significant effects of the infection compared with the group before the infection. Regarding gene expression of antimicrobial compounds, it was significantly higher in the 24 h post-infection group than in the infection group (*c3*, *mpo*, and *lyso*: *p* value < 0.001, *p* value < 0.01, and *p* value < 0.001, respectively).

## 4. Discussion

In this study, we evaluated the impact of an infection by *Aeromonas salmonicida achromogenes* can induce on microbiota level of different tissues through 16S rRNA sequencing.

Because the mucus layer is the primary interface between the environment and the host, fish have developed lymphoid tissues associated with this mucus layer. Although some studies highlight the impact of parasitic infections on salmonid alpha diversity [13,39], in this work no significant difference due to bacterial infection was observed. Reid et al. [39] also studied the impact of another pathogen, the salmonid alphavirus (SAV) on the bacterial community of salmon skin where they did not detect significant differences due to high within-group variation. Those studies focused on the impact of parasites and viruses and, therefore, a bacterial infection could have differently affected bacterial communities, but this was not the case.

However, strong differences in diversity between sites in the bacterial community were observed, with higher diversities for water and gills. Some studies have been conducted to determine the diversity between different body sites in fish. Lowrey et al. [40] performed a topographic mapping of the different tissues of rainbow trout. They found that the skin microbiota has a higher species diversity than the gill microbiota, which is in contradiction with the present results. Another study conducted on Atlantic salmon *Salmo salar* [41] found high species diversity in the water and lower diversity for the skin microbiota with similar values to ours. By combining the richness (Chao1) and diversity (Shannon and Inverse Simpson) indices calculated in the present study, we can suggest that the composition of the different microbiota is highly diverse but dominated by a few different taxa.

In addition to differences in alpha diversity between body sites (higher in gills and lower for skin and fin), we revealed a change in β-diversity as well. We showed that the diversity between communities at different body sites are different from each other suggesting that all these sites represent specialized ecological niches with their own community. Moreover, an impact of bacterial infection on the composition of the different communities was highlighted but at different times depending on the body site. The early changes in the gill bacterial community may be an indication that this body site could be a major site of contact and entry for *A. salmonicida* during infection. The gills are a highly vascularized organ due to their role in the uptake of dissolved oxygen from the water and, therefore, may be targeted by the pathogen for infection [24]. This is because the gill surface is primarily covered by a single layer of epithelium, unlike the skin, which has a much thicker layer of epithelium [42]. This primary infection of the gills was observed for *Yersinia ruckeri* in rainbow trout [17]. Ohtani et al. [18] showed similar results using IHC and optical projection tomography but also proposed the skin as a secondary entryway for the bacterial infection. These results are in accordance with our findings showing that the microbiota of the gills is the first impacted when facing bacterial infection.

Moreover, the bacterial community in the water was not altered by the changes in fish microbiota due to the bacterial infection and the potential release of the pathogen in the water. No differences in microbiota between body sites were shown, but rather an effect of bacterial infection alone. This suggests that the bacterial compositions between body sites were similar but not as abundant. It also suggests that bacterial infection caused some bacteria that were not present in the microbiota initially to appear and others to disappear. Therefore, we can state that the bacterial infection caused by *A. salmonicida achromogenes* resulted in dysbiosis. Bottleneck effects were also observed, caused by the bacterial infection, leading to a reduction in inter-individual bacterial diversity in each microbiota. This loss of bacterial diversity may be the starting point for infection by opportunistic pathogens, as they have much more space to grow and expand. Overall, the core microbiota of the gills and the skin are largely dominated by the genus *polynucleobacter* which is a bacterium of the *Proteobacteria* phylum and is known to be highly present in the freshwater environment [43]. *Polynucleobacter* is also abundant in fin microbiota but less dominant since bacteria from the *Comamonodaceae* family (*Proteobacteria* phylum) take a bigger part. Interestingly, we first observed an increase in the Proteobacteria phylum 6 h after bacterial infection. However, the relative abundance drastically dropped 72 h after infection. In a recent study, Zhan et al. showed that rainbow trout infected by infectious hematopoietic necrosis virus also exhibited a decrease in Proteobacteria accompanied by an increase in Actinobacteria [44].

Here, the decrease in *Proteobacteria* relative abundance 72 h after the bath infection was mainly due to the decrease in bacteria from the *polynucleobacter* genus and *Comamonadaceae* family. Although the proteobacteria phylum encompasses many different bacteria with different lifestyles, it is recognized that a decrease in the proportion of proteobacteria can lead to dysbiosis [45,46]. The core microbiota of the caudal fin is mainly dominated by *Comamonadaceae* and *Flavobacteriaceae.* This decrease is coupled with a large increase in *Flavobacterium* and *Pseudomonas* genus that are genus well-known to contain some opportunistic pathogens such as *Flavobacterium psychrophilum*, *columnare*, *Pseudomonas putida*, *aeruginosa*, and *fluorescens* [47,48]. Moreover, an increase in *Aeromonas* genus was observed in the core microbiota and concomitantly with the expression of *vapa* gene in the head kidneys that confirmed the presence and the virulence of *A. salmonicida achromogenes*. These observations are confirmed by a previous study that highlighted that *A. salmonicida* infection is associated with an increase in opportunistic pathogens on the skin of Atlantic salmon [49]. This expansion of opportunistic pathogens in microbiota was described following infection by a virus [39] but also following a parasitic infection [13]. Moreover, we observed an increase in *O. Undibacterium* in every microbiota after the bacterial challenge. *Undibacterium* has been described as a stress biomarker in pH stress in Tambaqui [45] but also has been reported to increase in the rat feces with experimental autoimmune encephalomyelitis (EAE) [50]. Therefore, *Undibacterium* may be an interesting candidate to follow the stress status of the fish while studying the microbiota.

Although the impact of bacterial infection seems less obvious on the caudal fin and skin microbiota networks, the changes observed in the gill microbiota correlation network are less equivocal. Indeed, the dysbiosis observed is not only due to an increase in opportunistic pathogens but also to changes in the relationships within the community. Indeed, many bacteria belonging to the different core microbiota increased or decreased after infection with *A. salmonicida achromogenes*. These observations were reinforced by the dysbiosis ratios as half of the bacterial network was modified showing a huge effect of *A. salmonicida achromogenes*. It has already been shown that stress [16] or pathogen infection [44,46] induce multiple modifications in the relationship between bacterial populations. Understanding the dynamics within a bacterial community while the host is undergoing infection is crucial for the development of pro/pre/synbiotics in aquaculture. Tax4Fun was used to assess the gene functions associated with the bacterial community of the different microbiota locations. First, this method confirmed that the caudal fin, gills, and skin microbiota as well as bacterial community of the water consist of specialized ecological niches. The most predominant differences are between the functional profiles of the skin with higher percentage of amino acids metabolism and energy metabolism compared with the fin functional profile which has a higher percentage of carbohydrate metabolism. However, unlike previous analyses, only the gill microbiota is impacted by the bacterial challenge while other organ microbiotas are not modified. Therefore, these microbiota modifications are in relation with the observed immune status modifications. The expression levels of pro-inflammatory cytokines (il-1β and tnf-α) increased 6 h after bacterial infection, indicating that the fish is in a state of stress and is trying to cope with the bacterial infection, but it never returned during the duration of the experiment to control levels. This suggests that the bacterial infection induced strong inflammatory responses and is combined with the inability of the fish to come out of this state by producing anti-inflammatory cytokines (il-10 and tgf-β). This hypothesis is supported by another study [51] that highlighted a strong expression of *il-1β* and *tnf-α* early in the infection by *A. salmonicida achromogenes*. Schwenteit et al. [52] showed an increase in pro-inflammatory cytokines in the head kidney of the Artic Charr (*Salvelinus alpinus, L*), which is a related species to rainbow trout, 3 days post-infection by *A. salmonicida. Il-1β* and *tnf-α* are from two different cytokine families which have overlapping functions, notably by the induction of other cytokines from activated lymphocytes [53].

About the regulatory mechanisms, we did not observe an effect of the bacterial infection on the anti-inflammatory cytokines in the head kidney. Brietzke et al. [51] highlighted a similar result to ours where *tgf-β* remained unmodified after the bacterial infection by *A. salmonicida*. Interestingly, *A. salmonicida* exerts strong immune suppressive effects through *il-10* expression from leukocytes of the head kidney culture cell [54]. This strong immunosuppression through *il-10* over-expression was not confirmed by our in vivo experiment.

The expression of the antimicrobial compounds (*c3*, *mpo*, and *lysozyme*) mRNA showed a similar pattern, meaning a delayed answer after 24 h post-infection and then a return to the control state except for the *mpo*.

## 5. Conclusions

In this study, a bath infection with *A. salmonicida achromogenes* significantly altered the gill, skin, and caudal fin microbiota in rainbow trout. In particular, beta diversity measurements showed that the gill microbiota was altered only 6 h after infection, subsequently affecting the skin and caudal fin microbiota, indicating that the gills may be the gateway to this infection. Negative binomial GLMs, associated correlation networks and dysbiosis ratios clearly revealed significant changes in the different microbiota. Along with the dysbiosis, the immune system attempted to cope with the infection by producing pro-inflammatory cytokines, resulting in high stress levels in the fish. A delayed response in the expression of antimicrobial compounds was also observed 24 h after infection. Therefore, we demonstrated that furunculosis not only damages the immune system but also induces dysbiosis in a tissue- and time-dependent manner post-infection.

## Figures and Tables

**Figure 1 microorganisms-11-00539-f001:**
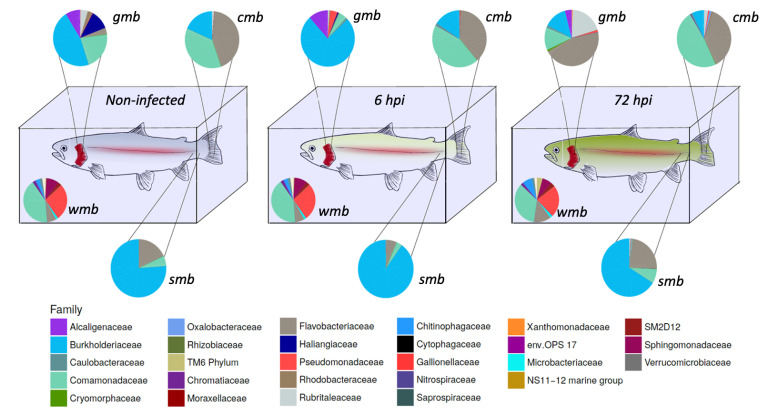
Bacterial composition of the core microbiota from different epithelia (gills: *gmb*; skin: *smb*; and caudal fin: *cmb*) and water before (before the infection: blue fish on the left side) and after bacterial challenge (6 h post-infection: yellow fish in the middle; and 72 h post-infection: green fish on the right side) with *Aeromonas salmonicida achromogenes*. OTUs were merged by family in order to keep the maximum information while reducing the noise from OTU number and are indicated under the draw.

**Figure 2 microorganisms-11-00539-f002:**
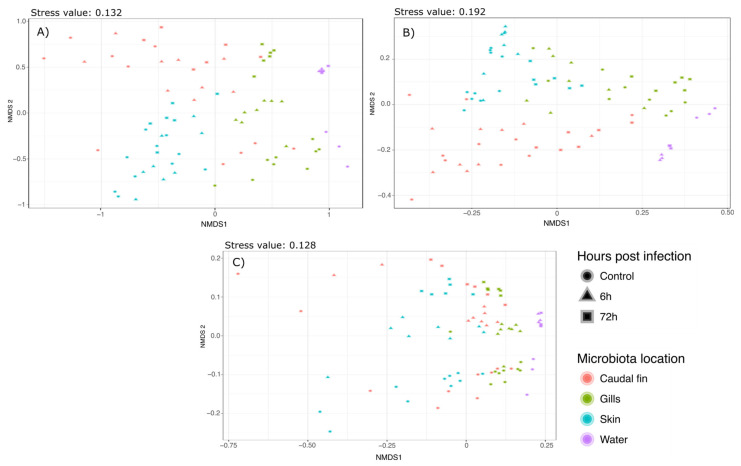
Nonmetric multidimensional scaling clustering microbial communities based on (**A**) Bray–Curtis, (**B**) Weighted, and (**C**) Unweighted Unifrac distances. The samples were clustered following this: circles represent the control condition, triangles represent the 6 h post-infection condition, and the squares represent the 72 h post-infection condition. Samples were also colored by the microbiota location (blue: skin microbiota; red: fin microbiota; green: gills microbiota; and purple: bacterial community of the water).

**Figure 3 microorganisms-11-00539-f003:**
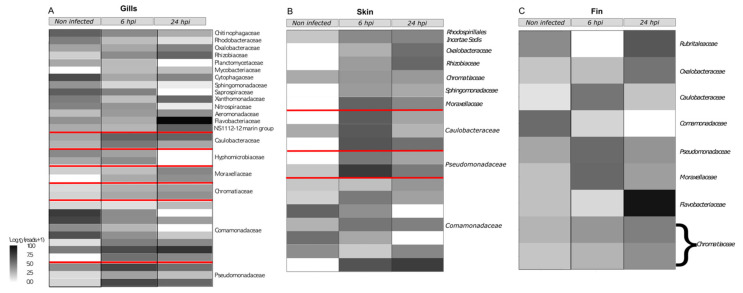
Heatmaps representing gills (**A**), skin (**B**), and fin (**C**) negative binomial GLM associated correlated network (Appendix A). Red lines delimitate OTUs representing the same bacterial genus.

**Figure 4 microorganisms-11-00539-f004:**
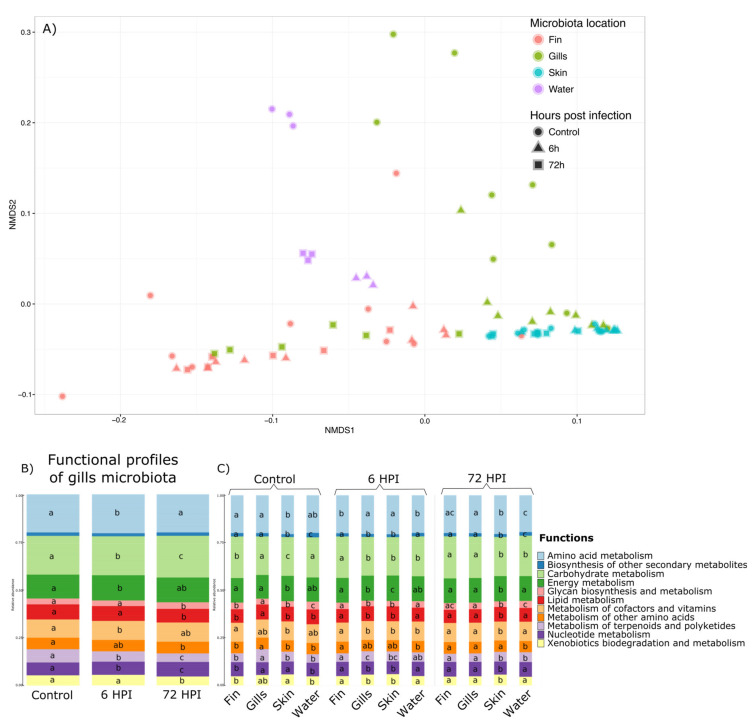
Tax4Fun analysis for the prediction of functional profiles of the rainbow trout microbiota. (**A**) Nonmetric multidimensional scaling clustering microbial communities using Bray–Curtis distance of 6324 KEGG orthology pathways. The samples were clustered following this: circles represent the group before the infection condition, triangles represent the 6 h post-infection condition, and the squares represent the 72 h post-infection condition. Samples were also colored by the microbiota location (blue: skin microbiota; red: fin microbiota; green: gills microbiota; and purple: bacterial community of the water). (**B**) Bar plots of the functional profiles of the gill microbiota in the group before the infection, 6 and 72 hpi. Letters represent the statistical differences observed by GLM analyses. (**C**) Bar plots of the functional profiles of the different microbiota in the group before the infection, 6 and 72 hpi. Letters represent the statistical differences observed by GLM analyses.

**Figure 5 microorganisms-11-00539-f005:**
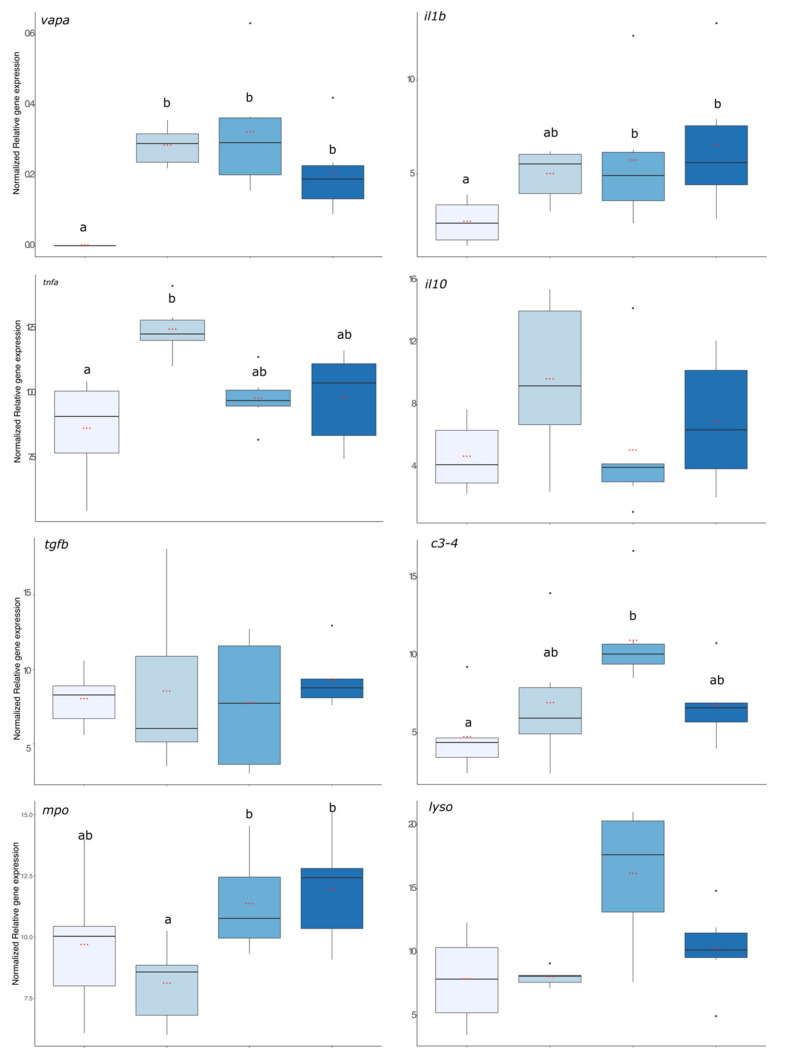
Evaluation of the immune status of rainbow trout through qPCR on head kidney in group before the infection and after bacterial infection (6, 24, and 72 hpi). The graphs represent *vapa, il1b, tnfa*, *il10, tgfβ*, *c3-4*, *mpo*, and *lyso* mRNA expressions. Letters (a and b) on the top of the plot represent the significance between groups and the three red dots in the boxplot represent the mean of the associated group. The black dot represents outlier fish.

**Table 1 microorganisms-11-00539-t001:** Sequence and qPCR efficiency for each primer used for gene expression analysis by real-time RT-PCR.

Gene	GeneBank Accession #	Sequence	qPCR Efficiency
*18s*	FJ710873.1	Fwd: 5′-TGAGCAATAACAGGTCTGTG-3′ Rv: 5′-GGGCAGGGACTTAATCAA-3′	101.7%
*ß-actin*	NM_001124235.1	Fwd:5′-GGACTTTGAGCAGGAGATGG-3′ Rv: 5′-ATGATGGAGTTGTAGGTGGTCT-3′	91%
*il-10*	NM_001245099.1	Fwd: 5′-CCGCCATGAACAACAGAACA-3′ Rv: 5′-TCCTGCATTGGACGATCTCT-3′	110.1%
*tgfß*	NM_001281366.1	Fwd: 5′-GCCAAGGAGGTCCACAAGTT-3′ Rv: 5′-GTGGTTTTGATGAGCAGGCG-3′	99.5%
*tnfα*	NM_001124374.1	Fwd: 5′-TGGTGCAAAAGATACCCACA-3′ Rv: 5′-GCACTGTGTCAGCGGTAAGA-3′	101.9%
*il1b*	NM_001124347.2	Fwd: 5′-CGTCACTGACTCTGAGAACAAGT-3′ Rv: 5′-TGGCGTGCAGCTCCATAG-3′	110%
*mpo*	GBTD01119227	Fwd: 5′-ATCCACACGGGCATCACCTG-3′ Rv: 5′-GCAGAGTCACCAATGACACCA-3′	97.5%
*c3-4*	L24433.1	Fwd: 5′-GAGATGGCCTCCAAGAAGATAGAA-3′ Rv: 5′-ACCGCATGTACGCATCATCA-3′	98.96%
*lyzo*	NM_001124716.1	Fwd:5′-TGCCTGTCAAAATGGGAGTC-3′ Rv: 5′-CAGCGGATACCACAGACGTT-3′	97.2%
*vapa*	AJ749879	Fwd: 5′-ATTAGCCCGAACGACAACAC-3′ Rv: 5′-CCAACACAATGAAACCGTTG-3′	98.25%

**Table 2 microorganisms-11-00539-t002:** Alpha diversity measurements indexes ± standard error mean (SEM) across microbiota location using GLM with Gaussian distribution.

	Fin	Gills	Skin	Water
Shannon	2.449 ± 0.192	3.578 ± 0.175	1.383 ± 0.107	5.094 ± 0.277
Inverse_Simpson	5.455 ± 1.036	10.45 ± 2.009	1.981 ± 0.159	28.925 ± 5.953
Chao1	828.683 ± 325.819	2533.665 ± 1909.024	638.504 ± 214.579	1407.951 ± 284.707

## Data Availability

Please contact author for data requests.

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
