# Peer review of "The Pathogen Aeromonas salmonicida achromogenes Induces Fast Immune and Microbiota Modifications in Rainbow Trout"

_microorganisms, 2023, doi:10.3390/microorganisms11020539_

Round 1

Reviewer 1 Report

The article is well-written and easy to follow. The material and methods are well described. However, there is a major inconsistency regarding the A. salmoncida species described and utilized here. A. salmonicida achromogenes is characterized by the absence of VapA. However, the authors used vapA genes in the qPCR assay, suggesting the presence of this protein array in the A. salmonicida strain used. At this point, the author should include a full characterization of the strain or a reference that fully describes the microbiological characteristics of this strain, including its classification. Also, it is not clear to me whether this strain is virulent. Please, provide evidence that a virulent strain of A. salmonicida has been used. Please, provide the genome accession number if possible. 

Please, make sure that all gene and species names are italicized in the manuscript (e.g., the first sentence of the conclusions).  

Please, use the term head-kidney instead of kidney. 

Table 1, modify VapA to vapA.

Figures 3, 4, and 5 need to be improved. The names are too small, and cannot be read. Also, the figures are not very intuitive and complex to analyze or visualize. Perhaps the should be moved to the supplementary figures.

Include the Ct values in a supplementary table. Please, include the selection of the normalizers and the G Norm value for these normalizers. 

Figure 7, please include the sample time point he the X-axis. Also, indicate the tissue that was studied. 

Please, look at the nomenclature (e.g., 6h, 24h, and 72hpi should be modified to  6, 24, 74 hpi). 

Please, make sure that all the figures are intuitive by adding labels that allow seeing the time points and the tissue sampled. Figure 1 is great, but needs basic labels (non-infected, 6 hpi, 72 hpi). 

Why did the authors not include the gut?

Why was the qPCR conducted in the head kidney and not in the gills?

Author Response

The article is well-written and easy to follow. The material and methods are well described. However, there is a major inconsistency regarding the A. salmoncida species described and utilized here. A. salmonicida achromogenes is characterized by the absence of VapA. However, the authors used vapA genes in the qPCR assay, suggesting the presence of this protein array in the A. salmonicida strain used. At this point, the author should include a full characterization of the strain or a reference that fully describes the microbiological characteristics of this strain, including its classification. Also, it is not clear to me whether this strain is virulent. Please, provide evidence that a virulent strain of A. salmonicida has been used. Please, provide the genome accession number if possible. 

We thank the reviewer for its comments and would like to remove any fear he could have on the strain used in our paper. The species used in this paper was initially isolated from infected fish in our laboratory. The strain was sent for identification to an INRA center that confirmed this strain was A. salmonicida achromogenes using Biolog system. It has been deposited in the BCCM collection (Belgian coordinated collection of microorganisms) with code LMG p-31558 but no genome accession is available for this strain yet. Regarding vapA, Gulla et al. (2016) used vapA gene to differentiate A. salmonicida subspecies and demonstrated that all acknowledged A. salmonicida subspecies used in their study, including A. salmonicida achromogenes formed a distinct and exclusive A-layer type, confirming the presence of this gene in the A. salmonicida achromogenes. Only one subspecies, ssp. pectinolytica, did not exhibit an A-layer. In addition, several sequences of vapA gene are available on the ncbi nucleotide bank (see accession number: AM937255.1, AJ749879.1). Regarding the virulence, the gene expression analysis showed an increase of vapA already 6h after exposure in trout head kidney attesting of the presence of A. salmonicida achromogenes in the trout organism and the expression of its virulence gene. Moreover, several clinical signs of furunculosis were observed which is the sign of an infection by this pathogen. Finally, this same strain was used in another experiment where a preliminary test infection, including various bacterial doses was performed to determine the LD50 CFU of the targeted rainbow trout population (LD50 = 3.1 × 107 CFU/100 g fish body weight) (Cornet et al. 2021).

We added information about this strain in the Material and methods.

Gulla, S.; Lund, V.; Kristoffersen, A.B.; Sørum, H.; Colquhoun, D.J. VapA (A-Layer) Typing Differentiates Aeromonas Salmonicida Subspecies and Identifies a Number of Previously Undescribed Subtypes. J. Fish Dis. 2016, 39, 329–342, doi:10.1111/jfd.12367.

Cornet, V.; Khuyen, T.D.; Mandiki, S.N.M.; Betoulle, S.; Bossier, P.; Reyes-López, F.E.; Tort, L.; Kestemont, P. GAS1: A New β-Glucan Immunostimulant Candidate to Increase Rainbow Trout (Oncorhynchus Mykiss) Resistance to Bacterial Infections With Aeromonas Salmonicida Achromogenes. Front. Immunol. 2021, 12, 1–16, doi:10.3389/fimmu.2021.693613.

Please, make sure that all gene and species names are italicized in the manuscript (e.g., the first sentence of the conclusions).  

This was revised in the text.

Please, use the term head-kidney instead of kidney. 

This was changed in the text.

Table 1, modify VapA to vapa.

This was changed in the text.

Figures 3, 4, and 5 need to be improved. The names are too small and cannot be read. Also, the figures are not very intuitive and complex to analyze or visualize. Perhaps the should be moved to the supplementary figures.

The figures were improved to allow an easier reading. In addition, we followed the advice of the reviewer and moved it in supplementary figures. These were replaced by figure 3 that only shows the heatmap of the microbiota of the three epithelia.

Include the Ct values in a supplementary table. Please, include the selection of the normalizers and the G Norm value for these normalizers. 

For gene expression, we used two reference genes: b-actin that was proved to be stable in the same context in a previous experiment (Cornet et al. 2018) and 18S RNA is highly recommended as internal standards for mRNA quantification study because mRNA variations are weak in comparison and cannot highly modify the total RNA level (Thellin 1999). Bestkeeper results showed a coefficient of correlation of 0,926 (p<0.001), SD of 0,19 and 0,941 (p<0.001) SD of 0,37 for 18S RNA and  b-actin respectively. The Bestkeeper values suggested was the geometric mean of both gene as we did in our experiment. A table including Ct values for each gene was added in the supplemental material.

Cornet, V.; Ouaach, A.; Mandiki, S.N.M.; Flamion, E.; Ferain, A.; Van Larebeke, M.; Lemaire, B.; Reyes López, F.E.; Tort, L.; Larondelle, Y.; et al. Environmentally Realistic Concentration of Cadmium Combined with Polyunsaturated Fatty Acids Enriched Diets Modulated Non-Specific Immunity in Rainbow Trout. Aquat. Toxicol. 2018, 196, 104–116, doi:10.1016/j.aquatox.2018.01.012.

Thellin, O.; Zorzi, W.; Lakaye, B.; De Borman, B.; Coumans, B.; Hennen, G.; Grisar, T.; Igout, A.; Heinen, E. Housekeeping Genes as Internal Standards: Use and Limits. J. Biotechnol. 1999, 75, 291–295, doi:10.1016/S0168-1656(99)00163-7.

Figure 7, please include the sample time point he the X-axis. Also, indicate the tissue that was studied. 

The sample time was included in the X-axis and the tissue was indicated.

Please, look at the nomenclature (e.g., 6h, 24h, and 72hpi should be modified to 6, 24, 74 hpi). 

The nomenclature was checked and changed as asked.

Please, make sure that all the figures are intuitive by adding labels that allow seeing the time points and the tissue sampled. Figure 1 is great, but needs basic labels (non-infected, 6 hpi, 72 hpi). 

Several labels were applied in Figure 1 to allow an easier understanding.

Why did the authors not include the gut?

In this study, the focus was made on epithelia and their associated microbiota that are in direct contact with the water. The addition of the gut would have made the experimental design more complex and comparison with other microbiota would have been complicated due to the presence of food in their digestive tract and their inner location.

Why was the qPCR conducted in the head kidney and not in the gills?

The head kidney is a major organ of the immune system, and the aim was to evaluate if the fish immune system was efficiently triggered by the bacterial challenge. The analysis of the organ inside the fish that is not supposed to contain A. salmonicida achromogenes also allowed us to ensure the presence and virulence of this strain in the fish organism. In addition, the gill mucus transcriptome was performed in Illumina along with the microbiota metatranscriptome and should be the subject of a separate paper.

Reviewer 2 Report

The title "The pathogen Aeromonas salmonicida achromogenes induces fast immune and microbiota modifications in rainbow trout " has been reviewed. The taxonomic composition and structure of the infected microflora were evaluated, and the immune status of the fish was detected. I think it makes sense for the salmonid aquaculture and immunity, it may be published in microorganisms after some modifications.

Abstract:

1) The abstract of this article looks ok, but some descriptions are redundant, and I hope to modify them.

2) It is suggested to write the keywords after the abstract.

Introduction:

1) Page 1, “Boutin et al.  [8] showed that indigenous bacteria from unstressed brook charr (Salvelinus fontinalis) skin can exert antagonistic effect against Flavobacterium infection.” Incorrect citation format.

Materials and methods:

1) Page 3, “Using the same protocol than for fish euthanized before bacterial infection, ten fish were randomly sampled at 6 and 24 hours post infection (hpi); and 6 fish were sampled at 72 and 168 hpi. ” Why is the number of random samples different?

2) Page 3, “ A control group that was not exposed to the bacteria was also conducted on three rainbow trouts (skin and gill mucus) in order to follow the changes that might occur naturally in the different microbiotas within 72 hours. ” What about the sample 168 h post-infection? Need to be described clearly.

3) Page 4, “The immune response of fish throughout the bacterial infection was assessed using immune gene expression from the kidneys. ” To describe accurately, write about the head kidneys.

4) Page 5, “Two housekeeping genes (β-actin and 18S rRNA) were used to normalized the expression of the target genes.” What is the reason for choosing two housekeeping genes?

Result:

1) Page 5, “Several signs of hemorrhages at the base of the fins,  small ulcers on the side of the fish and mortalities were observed.” It is suggested to supplement the pictures of the clinical symptoms of fish.

2) It is suggested that the tables and pictures of the results should be clearly marked and written uniformly.

Discussion:

There is no need to repeat the results in the discussion.

Figure:

1) Figures 4 and 5 suggest a little more clarity.

2) What does the black dot in Figure 7 represent? And the name of the gene in the picture should be consistent with that of the paper. What is 8.a in the note in Figure 7? Please check.

References:

Delete the blank number 52.

Author Response

Abstract:

1) The abstract of this article looks ok, but some descriptions are redundant, and I hope to modify them.

This abstract was modified to avoid redundancy.

2) It is suggested to write the keywords after the abstract.

The keywords were added just after the abstract.

Introduction:

1) Page 1, “Boutin et al.  [8] showed that indigenous bacteria from unstressed brook charr (Salvelinus fontinalis) skin can exert antagonistic effect against Flavobacterium infection.” Incorrect citation format.

The number was placed at the end of the sentence.

Materials and methods:

1) Page 3, “Using the same protocol than for fish euthanized before bacterial infection, ten fish were randomly sampled at 6 and 24 hours post infection (hpi); and 6 fish were sampled at 72 and 168 hpi. ” Why is the number of random samples different?

Unfortunately, we observed some mortalities and moribund fish at 72 hpi, it was just possible to sample 6 fish at those timepoints.

2) Page 3, “ A control group that was not exposed to the bacteria was also conducted on three rainbow trouts (skin and gill mucus) in order to follow the changes that might occur naturally in the different microbiotas within 72 hours. ” What about the sample 168 h post-infection? Need to be described clearly.

Due to budget limitation, this timepoint was not analyzed in this study. Its mention was not supposed to appear here and was removed from the material and methods.

3) Page 4, “The immune response of fish throughout the bacterial infection was assessed using immune gene expression from the kidneys. ” To describe accurately, write about the head kidneys.

Kidney was replaced by head kidney in the text.

4) Page 5, “Two housekeeping genes (β-actin and 18S rRNA) were used to normalized the expression of the target genes.” What is the reason for choosing two housekeeping genes?

The use of stable housekeeping genes among biological replicates and conditions is essential to consolidate the results as several technical manipulation could induce variability in the amount of RNA loaded in the qPCR plate. Since few years, several studies advised the use of at least two housekeeping genes to avoid substantial errors (Bustin et al. 2008; Nicot et al. 2005).

Bustin, S.A., Benes, V., Garson, J.A., Hellemans, J., Huggett, J., Kubista, M., Mueller, R., Nolan, T., Pfaffl, M.W., Shipley, G.L., Vandesompele, J., 2009. The MIQE guidelines: minimum information for publication of quantitative real-time PCR experiments.Clin. Chem. https://doi.org/10.1373/clinchem.2008.112797

Nicot, N., Hausman, J.F., Hoffmann, L., Evers, D., 2005. Housekeeping gene selection for real-time RT-PCR normalization in potato during biotic and abiotic stress. J. Exp.Bot. 56 (421), 2907–2914. https://doi.org/10.1093/jxb/eri285.

Result:

1) Page 5, “Several signs of hemorrhages at the base of the fins, small ulcers on the side of the fish and mortalities were observed.” It is suggested to supplement the pictures of the clinical symptoms of fish.

Unfortunately, the fish were kept in a confined laboratory (Lv.2) and it is forbidden to bring camera or phone inside this room. However, we added a reference that explains and shows the different symptoms of furunculosis.

Baset, A. Status of Furunculosis in Fish Fauna. In Bacterial Fish Diseases; Elsevier, 2022; pp. 257–267.

2) It is suggested that the tables and pictures of the results should be clearly marked and written uniformly.

This was changed according to the reviewer’s comment and are marked in red in the text.

Discussion:

There is no need to repeat the results in the discussion.

The discussion was changed to avoid the repetition of results in it.

Figure:

1) Figures 4 and 5 suggest a little more clarity.

Those figures were adapted. Only the heatmaps were kept and gathered to form the figure 3. On the recommendation of the reviewer, those figures were placed in the supplementary material and changed to be clearer (increase of the font size).

2) What does the black dot in Figure 7 represent? And the name of the gene in the picture should be consistent with that of the paper. What is 8.a in the note in Figure 7? Please check.

The black dots represent outlier values. It was added in the legend.

The name of the genes was checked to be consistent between the text and the figure. The legend was also changed as the figure 8.a was not existing in this paper version.

References:

Delete the blank number 52.

Unfortunately, the blank cannot be changed because it seems to be due to Mendeley paging. It will probably be removed during editing process.

Reviewer 3 Report

This manuscript provides an in-depth analysis of the surface microbiome of fish exposed to Aeromonas salmonicida, a very important fish pathogen. I believe the paper should ultimately be published after the authors make the revisions that I recommend below.

For context, my expertise is in fish pathology and infectious diseases, not microbiome analyses, and thus my recommendations are in line with this.

First, throughout the paper the authors state “infection”, but actually all they can claim based on the data presented is that the exposed group was “exposed”, and only perhaps infected.  Fish presented with clinical signs consistent with the disease, but the infection was not verified in these fish.  As they already have abundant DNA samples from the gills, etc., I request that the authors go back and perform a qPCR analysis on their samples using a specific test for A. salmonicida.  This way, they can compare their microbiome results with actual burden of the pathogen on a fish by fish basis.  This would really give them tools for much more powerful analyses and comparisons.  For example, at various places, the authors note increase of “Aeromonas”, but this may simply be an environmental species, not A. salmonicida.  See page 18 “Moreover, we also found an increase in Aeromonas, which may be the pathogen we used for the bathing infection”.

The authors mention that some fish became moribund and showed lesions.   Please provide data on how many fish and when they showed signs of disease.  Again, I strongly recommend analysis comparing clinically normal fish and these moribund fish, which we presume have furunculosis.  In other words, a very interesting analysis is missing and could be included by some additional analyses from samples that they already collected. Perhaps A. salmonicida abundance or moribidity strongly correlate with a microbiome profile.

Experimental Design.  It appears that there were not tank replicates for each time point.  This is important as the researchers hence cannot account for tank as a co-founding variable.  This should be stated, but also it is evident that there was no significant differences in the water microbiomes, so one could use this as justification that there likely no tank effects.

Author Response

First, throughout the paper the authors state “infection”, but actually all they can claim based on the data presented is that the exposed group was “exposed”, and only perhaps infected.  Fish presented with clinical signs consistent with the disease, but the infection was not verified in these fish.  As they already have abundant DNA samples from the gills, etc., I request that the authors go back and perform a qPCR analysis on their samples using a specific test for A. salmonicida.  This way, they can compare their microbiome results with actual burden of the pathogen on a fish by fish basis.  This would really give them tools for much more powerful analyses and comparisons.  For example, at various places, the authors note increase of “Aeromonas”, but this may simply be an environmental species, not A. salmonicida.  See page 18 “Moreover, we also found an increase in Aeromonas, which may be the pathogen we used for the bathing infection”.

The authors agreed that an infection can be stated only if there are proof of the entry and virulence of the bacteria. In this study, we didn’t used the DNA extracted from the gills and skin because it was coming from mucus swapping and we couldn’t have assessed that this was not coming for residual bacteria in the mucus. To prove the entry of A. salmonicida achromogenes, we sampled a lymphoid organ to evaluate its specific vapa gene expression. This gene assessed the presence of this bacteria in the fish but cannot be used to quantify the burden of the pathogen. It would have been very interesting to get the amount of this bacteria in head kidney, unfortunately all the sampling was used to extract RNA.

The authors mention that some fish became moribund and showed lesions. Please provide data on how many fish and when they showed signs of disease.  Again, I strongly recommend analysis comparing clinically normal fish and these moribund fish, which we presume have furunculosis.  In other words, a very interesting analysis is missing and could be included by some additional analyses from samples that they already collected. Perhaps A. salmonicida abundance or moribidity strongly correlate with a microbiome profile.

In this experiment, the moribund fish were excluded from the analysis (explaining why we only sampled 6 fish at 72 hpi). According to ethical rules, fish that had reached the human acceptable endpoints had to be euthanized. As all the fish did not reach those points at the same time, it was not possible for us to perform a synchronized sampling with moribund and “apparently healthy but still infected fish”. Moreover, here, we wanted to evaluate the modification of several microbiota in fish infected by a bacterial pathogen, that immune system was triggered and that in turn could potentially alter the mucosal immunity of skin and gills and thus the relationship with its core microbiota. Using moribund fish that couldn’t cope with this pathogen would have been interesting but useless to address this question.

Experimental Design.  It appears that there were not tank replicates for each time point.  This is important as the researchers hence cannot account for tank as a co-founding variable.  This should be stated, but also it is evident that there was no significant differences in the water microbiomes, so one could use this as justification that there likely no tank effects.

The authors thank the reviewer for this suggestion, this was added to the materials and methods section.

Round 2

Reviewer 2 Report

The title "The pathogen Aeromonas salmonicida achromogenes induces fast immune and microbiota modifications in rainbow trout" has been studied, it can be published in Microorganisms.

Some formatting and spelling changes are listed below:

(1) In the abstract“at 6 and 72h post-infection (hpi)”,A space needs to be added between numbers and units.

(2) In section 2.2. In vivo experiment: bacterial challenge and fish sampling“3.1 × 107 CFU”, the number is not superscribed and should be changed to“3.1 × 107 CFU”. And “6, 24 and 72 hpi post infection”, should write “hpi” or “h post-infection”.

(3) In section 2.5. Immune gene expression analysis“head-kidney” and “head kidney”, unified format. And “p<0.001”Need to be changed to italics, to“p<0.001”.

(4) In section 3.7. Evaluation of the immune status of the fish “from the 6h, 24h and 72hpi groups” changed to “from the 6, 24 and 72 hpi groups”. And “For tnfa, its expression increased significantly 6 hpi”. It should be “tnfα”, not “tnfa”.